# An Observational Study on Cephalometric Characteristics and Patterns Associated with the Prader–Willi Syndrome: A Structural Equation Modelling and Network Approach

Alin Viorel Istodor [1], Laura-Cristina Rusu [2,3,*], Gratiela Georgiana Noja [3,4], Alexandra Roi [2,3], Ciprian Roi [3,5], Emanuel Bratu [6,*], Georgiana Moise [7], Maria Puiu [8,9], Simona Sorina Farcas [3,8,9] and Nicoleta Ioana Andreescu [3,8,9]

[1] First Department of Surgery, Second Discipline of Surgical Semiology, "Victor Babes" University of Medicine and Pharmacy, Timisoara, 2 Eftimie Murgu Square, 300041 Timisoara, Romania; istodor.alin@umft.ro

[2] Department of Oral Pathology, "Victor Babes" University of Medicine and Pharmacy Timisoara, 2 Eftimie Murgu Sq., 300041 Timisoara, Romania; alexandra.moga@umft.ro

[3] Multidisciplinary Center for Research, Evaluation, Diagnosis and Therapies in Oral Medicine, Oral Pathology Department, "Victor Babes" University of Medicine and Pharmacy Timisoara, 300041 Timisoara, Romania; gratiela.noja@e-uvt.ro (G.G.N.); ciprian.roi@umft.ro (C.R.); farcas.simona@umft.ro (S.S.F.); andreescu.nicoleta@umft.ro (N.I.A.)

[4] Department of Marketing and International Economic Relations, Faculty of Economics and Business Administration, West University of Timisoara, 16 Pestalozzi Street, 300115 Timisoara, Romania

[5] Department of Anaesthesiology and Oral Surgery, "Victor Babes" University of Medicine and Pharmacy Timisoara, 2 Eftimie Murgu Sq., 300041 Timisoara, Romania

[6] Department of Implant Supported Restorations, "Victor Babes" University of Medicine and Pharmacy, Faculty of Dentistry, 2 Eftimie Murgu Sq., 300041 Timisoara, Romania

[7] Department of Clinical Pharmacology, "Victor Babes" University of Medicine and Pharmacy, "Pius Brinzeu", County Emergency Clinical Hospital Timisoara, 2 Eftimie Murgu Square, 300041 Timisoara, Romania; drgeorgianamoise@gmail.com

[8] Center of Genomic Medicine, Department of Microscopic Morphology—Genetics, University of Medicine and Pharmacy "Victor Babes", 300041 Timisoara, Romania; maria_puiu@umft.ro

[9] Regional Center of Medical Genetics Timis, Clinical Emergency Hospital for Children "Louis Turcanu", 300011 Timisoara, Romania

* Correspondence: laura.rusu@umft.ro (L.-C.R.); ebratu@umft.ro (E.B.)

**Abstract:** Examining specific patterns of major cranio-facial alterations through cephalometric measurements in order to improve the Prader–Willi (PWS) syndrome diagnostic poses a major challenge of identifying interlinkages between numerous credentials. These interactions can be captured through probabilistic models of conditional independence between heterogeneous variables. Our research included 18 subjects (aged 4 to 28 years) genetically diagnosed with Prader–Willi syndrome and a healthy control group (matched age and sex). A morphometric and cephalometric analysis was performed upon all the subjects in order to obtain the needed specific data. We have, therefore, firstly deployed several integrated Gaussian graphical models (GGMs) to capture the positive and negative partial correlations and the intensity of the connections between numerous credentials configured to determine specific cranio-facial characteristics of patients with PWS compared to others without this genetic disorder (case-control analysis). Afterwards, we applied structural equation modelling (SEM) with latent class analysis to assess the impact of these coordinates on the prevalence of the Prader–Willi diagnostic. We found that there are latent interactions of features affected by external variables, and the interlinkages are strapping particularly between cranial base (with an important role in craniofacial disharmonies) and facial heights, as important characteristic patterns in determining the Prader–Willi diagnostic, while the overall patterns are significantly different in PWS and the control group. These results impact the field by providing an enhanced comprehensive perspective on cephalometric characteristics and specific patterns associated with Prader–Willi syndrome that can be used as benchmarks in determining the diagnostic of this rare genetic disorder. Furthermore, the two innovative exploratory research tools applied in this paper are very useful to the craniofacial field to infer the connections/dependencies between variables (particularly biological variables and genes) on cephalometric characteristics and specific patterns associated with Prader–Willi syndrome.

**Keywords:** Prader-Willi; cranio-facial alterations; Gaussian graphical model; structural equation modelling; data integration; diagnostic

## 1. Introduction

Prader–Willi syndrome (PWS) is a complex, rare genetic disease, first described by Prader et al. [1] in 1956 and characterized by a neurobehavior disorder. The incidence of this rare genetic disease cannot be evaluated, but it is estimated to be 1 in 15,000–20,000 births [2]. Further chromosome analysis proved that patients diagnosed with Prader–Willi syndrome have a chromosomal deletion localized on 15q11-q13 [3,4]. Numerous studies focused in identifying the exact genomic imprinting mechanisms that target the 15q11-q13 chromosome [5,6], and besides the advancement made in genomics the exact molecular mechanism that occurs in Prader–Willi syndrome is not well known.

The genes that are imprinted are responsible for the RNA and the protein processing of hormones and neuroregulators. The alterations encountered at this level affect the endocrine function and hormone levels that are responsible for proper skeletal development and neuronal function [7–10]. The majority of the documented cases (65–70%) occurred due to the deletion in chromosome 15, as many as 20–30% of the cases were caused by a uniparental disomy (maternal) of the chromosome 15, and 2–5% were due to an imprinting center defect [11].

PWS is considered to be a multisystemic disorder that, starting from infancy, determines neonatal hypotonia, alteration of the skeletal development, dysmorphic features, cognitive impairment and behavioral problems, multiple endocrine alterations, and severe hyperphagia with obesity [10,12]. It appears that a dysregulation localized in the hypothalamus is responsible for this specific phenotype and the manifestations differ during life, becoming more evident through adulthood [13].

Often this syndrome has clinical manifestations similar to other diseases, a fact that implies a further accurate diagnosis. The alterations of the cranio-facial area require attention from the clinician in order to evaluate and provide proper guidance in the treatment.

Prader–Willi syndrome has a genetic and epigenetic involvement with an important impact upon normal development, needing a complex and multidisciplinary therapy.

The clinical characteristics associated with PWS are short stature, small hands and feet, hypogonadism and cognitive impairment [14]. Also, during the development, cranio-facial anomalies can be identified in this type of patient. Meaney et al. [15] conducted a study based on anthropometry in order to obtain accurate information about the cranio-facial development in Prader–Willi syndrome.

In this case, the use of the cephalometric radiography analysis offers a complete set of information related to the cranio-facial structures and their growth potential. This type of radiography gives the clinician the opportunity to evaluate changes in the bony architecture and dental structures related to these patients, facts that can improve their diagnosis and further treatment. The cephalometric analysis is based on the measurement between various soft tissue and dentoskeletal landmarks in order to assess the facial proportions. Several studies stated that the cranio-facial anomalies encountered in Prader–Willi syndrome are palpebral fissures, a vermilion border, a narrow bifrontal diameter, almond-shaped eyes, heavy soft tissue draping over the chin, down-turned mouth corners, strabismus, dolichocephaly and smaller cranial measurements [15]. Also, other existing reports described the cranio-facial morphology of PWS patients after cephalometric analysis and outlined the fact that there are certain features such as a short antero-posterior length of the maxilla and a short mandibular ramus often encountered in these patients [15,16].

For the evaluation of cranio-facial development, the cephalometric analysis proved to be an asset for pediatricians, dentist, oral surgeons and orthodontist, that should be used from early age to monitor, establishing the pattern of growth, evaluating the morpho-functional perturbances.

The aim of our study is to use the evidence from the network analysis deployed based on Gaussian graphical models (GGMs) and structural equation modeling (SEM) configured through the integration of variables related to the cranio-facial alterations found in Prader–Willi syndrome in order to capture the patterns and overall linkages between the measurement units from the two groups of patients studied (Prader–Willi subjects and control group).

In this perspective, a major edge of GGMs is "the ability to handle different types of variables", like the ones captured in our sample, since the indicators/variables encompassed by the empirical assessment have different measurement units (e.g., binary, multi-category).

Furthermore, as a complex multivariate analysis technique, often used in social sciences, SEM is applied in this medical research to measure and analyze the relationship between the observed (measurements performed on Prader–Willi syndrome patients and on the control sample) and latent variables. Therefore, the presence of Prader–Willi syndrome is captured as a latent variable that is being calculated from the other measured variables (indicators/measurements compiled in our dataset) that reflect specific craniofacial characteristics of the persons from which these measurements were taken.

SEM advances this research endeavor since it combines path analysis, factor analysis and regression, and hence it facilitates to specify multiple causal associations between our constructs, as a major advantage for craniofacial studies. Therefore, we were able to model conditional associations, namely the degree in which the variables are independent after conditioning on all other variables in the data set. This characteristic was essential in our empirical research because we configured 29 major cephalometric measures that capture, in a gradual frame, the main coordinates and patterns of Prader–Willi syndrome beyond the genetic disorder, another important advantage in the craniofacial field.

These specific coordinates were evaluated in their tight interdependence and sequential approach, as a complex network (performed in this paper through the GGMs) and through causal relationships (as enhanced by the SEM models designed to achieve the complicated model setup).

Main findings from the Gaussian graphical models have evidenced strong edges and partial correlations between all considered credentials (cephalometric measures). These parameters tend to have reduced dimensions in PWS compared to the control sample. SEM results entail that the coefficients associated with Sella–Nasion (SN) coordinates (measurements) are both negative and statistically significant in the case of the Prader–Willi sample, while for the control sample these coefficients are positive, being also very significant from a statistical point of view. These coefficients entail that an increase in the prevalence of PWS is associated with a significant change and variation in the Sella–Nasion values, but also in other cephalometric characteristics, as evidenced and further detailed in the Results section.

## 2. Materials and Methods

### 2.1. Data—Representative Sample and Measurement Units

This research is based on a case-control analysis, relying on two samples, one sample with 18 subjects with Prader–Willi syndrome and the second was a control sample with 18 subjects without this genetic disorder. We have compiled a large dataset with 29 indicators/measurements for all 36 subjects comprised in our analysis. Summary statistics are described in Table 1a,b. A detailed description of each variable is presented in the Appendix A, Table A1.

Each sample consists of 18 subjects (36 subjects in total), 7 males and 11 females, in each sample (14 males and 22 females in total, as entailed in Figure 1), having different ages ranging from 4 to 28 years, as shown in Figure 2.

The study group included patients for whom the clinical diagnosis of Prader–Willi syndrome was confirmed by FISH ("Fluorescent in situ Hybridization") analysis or by the methylation test. The cephalometric measurements were performed during a period of 4 months. The control group included a similar number of individual, matched by age and

sex with the patients form the study group. The control group included only individual with a straight profile and without previous orthodontic treatment.

The cephalometric radiographs were performed using the technique standardized with central teeth in occlusion, with relaxed lips. The patients were place with the sagittal plane parallel to the film and the earbuds slightly inserted into the external auditory canal for position stabilization during exposure. No radiographic corrections were made.

**Table 1.** (**a**) Summary statistics for the first sample comprising 18 Prader–Willi patients, (**b**) summary statistics for the control sample (18 subjects without the disease).

| (a) | | | | | |
|---|---|---|---|---|---|
| | **N** | **Mean** | **Sd** | **Min** | **Max** |
| SN | 18 | 75.83333 | 1.158697 | 73.84 | 78.22 |
| PTM-N | 18 | 51.36722 | 2.375254 | 47.15 | 53.94 |
| AR-PTM | 18 | 39.80333 | 3.095097 | 31.77 | 43.31 |
| SAr | 18 | 36.40722 | 2.750743 | 30.76 | 39.71 |
| NSAr | 18 | 118.1539 | 5.262845 | 109.65 | 126.52 |
| NSBa | 18 | 127.9272 | 4.531925 | 120.73 | 133.84 |
| N-A-PG | 18 | 6.463333 | 1.5576 | 4.43 | 9.22 |
| ANS-N | 18 | 53.02833 | 4.486369 | 48.92 | 60.81 |
| ANS-Me | 18 | 58.04833 | 4.10118 | 55.18 | 68.02 |
| N-Me | 18 | 111.4622 | 7.348106 | 105.02 | 128.83 |
| S-Go | 18 | 71.94833 | 8.449223 | 65.23 | 85.98 |
| SN-MP | 18 | 34.42056 | 1.398527 | 33.12 | 39.15 |
| FH-MP | 18 | 25.56389 | 2.428631 | 23.28 | 31.54 |
| Co-A | 18 | 89.36333 | 2.57655 | 85.26 | 92.47 |
| ANPog | 18 | 3.172778 | 3.092271 | −8.65 | 5.72 |
| SNA | 18 | 83.85167 | 1.831523 | 78.95 | 86.52 |
| SNB | 18 | 80.11778 | 1.816926 | 75.79 | 82.42 |
| ANB | 18 | 3.730556 | 0.7057868 | 2.89 | 5.74 |
| SArGo | 18 | 146.2617 | 6.500576 | 137.51 | 159.19 |
| U1-L1 | 18 | 138.9233 | 4.288158 | 129.27 | 147.82 |
| ANS-PNS | 18 | 55.48167 | 2.146248 | 53.52 | 58.91 |
| Ar-Go | 18 | 39.53444 | 4.258378 | 35.89 | 49.15 |
| Go-Pog | 18 | 80.60833 | 3.458699 | 74.18 | 87.52 |
| Co-Gn | 18 | 111.2456 | 5.011898 | 106.14 | 118.97 |
| B-Pog | 18 | 7.232778 | 1.473267 | 4.67 | 9.75 |
| ArGoGn | 18 | 124.6311 | 2.364735 | 121.95 | 132.26 |
| *N* | 18 | | | | |
| (b) | | | | | |
| | **N** | **Mean** | **Sd** | **Min** | **Max** |
| SN | 18 | 70.19833 | 3.893115 | 63.23 | 77.32 |
| PTM-N | 18 | 52.11556 | 1.050738 | 50.28 | 53.46 |
| AR-PTM | 18 | 37.20333 | 0.8617149 | 35.9 | 38.45 |
| SAr | 18 | 33.22944 | 1.340042 | 31.25 | 36.04 |
| NSAr | 18 | 123.0789 | 0.5853593 | 122.25 | 123.93 |
| NSBa | 18 | 130.6839 | 1.605671 | 128.12 | 133.56 |
| N-A-PG | 18 | 5.203333 | 1.134165 | 3.53 | 7.23 |
| ANS-N | 18 | 54.25111 | 1.308829 | 52.54 | 56.35 |
| ANS-Me | 18 | 61.36778 | 1.744421 | 58.01 | 64.21 |
| N-Me | 18 | 115.5078 | 3.07832 | 110.86 | 120.56 |
| S-Go | 18 | 73.60722 | 6.495733 | 65.02 | 83.98 |
| SN-MP | 18 | 33.96944 | 1.104602 | 32.12 | 35.67 |
| FH-MP | 18 | 28.79222 | 0.7744261 | 27.45 | 30.56 |
| Co-A | 18 | 86.21722 | 1.84672 | 83.52 | 88.9 |
| ANPog | 18 | 2.386667 | 1.388444 | .34 | 4.24 |
| SNA | 18 | 81.83611 | 0.6042703 | 81.12 | 82.85 |
| SNB | 18 | 79.16556 | 1.225226 | 77.23 | 81.09 |

**Table 1.** *Cont.*

| | (b) | | | | |
|---|---|---|---|---|---|
| | **N** | **Mean** | **Sd** | **Min** | **Max** |
| ANB | 18 | 2.637222 | 0.7130614 | 1.56 | 3.89 |
| SArGo | 18 | 142.9033 | 2.160136 | 139.41 | 145.89 |
| U1-L1 | 18 | 126.3667 | 2.913298 | 122.85 | 130.56 |
| ANS-PNS | 18 | 55.21111 | 2.278048 | 52.24 | 59.11 |
| Ar-Go | 18 | 44.18667 | 3.546436 | 39.56 | 49.96 |
| Go-Pog | 18 | 82.20111 | 2.778252 | 79.12 | 87.23 |
| Co-Gn | 18 | 112.6739 | 4.123416 | 107.36 | 119.56 |
| B-Pog | 18 | 8.666111 | 0.4311973 | 7.93 | 9.35 |
| ArGoGn | 18 | 127.1172 | 4.059518 | 121.36 | 132.87 |
| *N* | 18 | | | | |

Source: Authors' contribution in Stata 16.

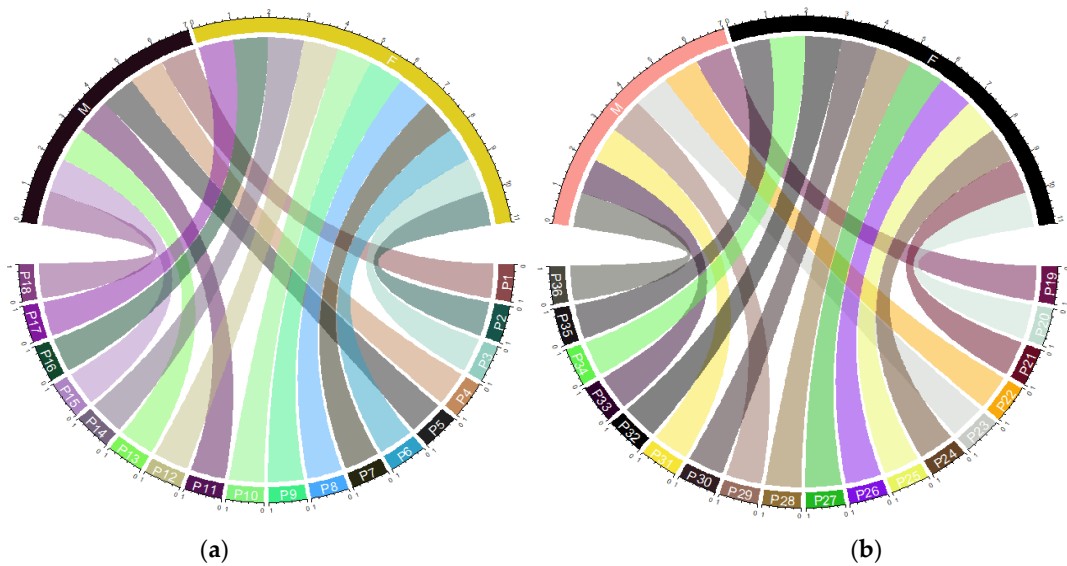

**Figure 1.** Sample representation by sex: (**a**) Prader–Willi sample; (**b**) control sample. Source: authors' contribution in R version 3.6.3.

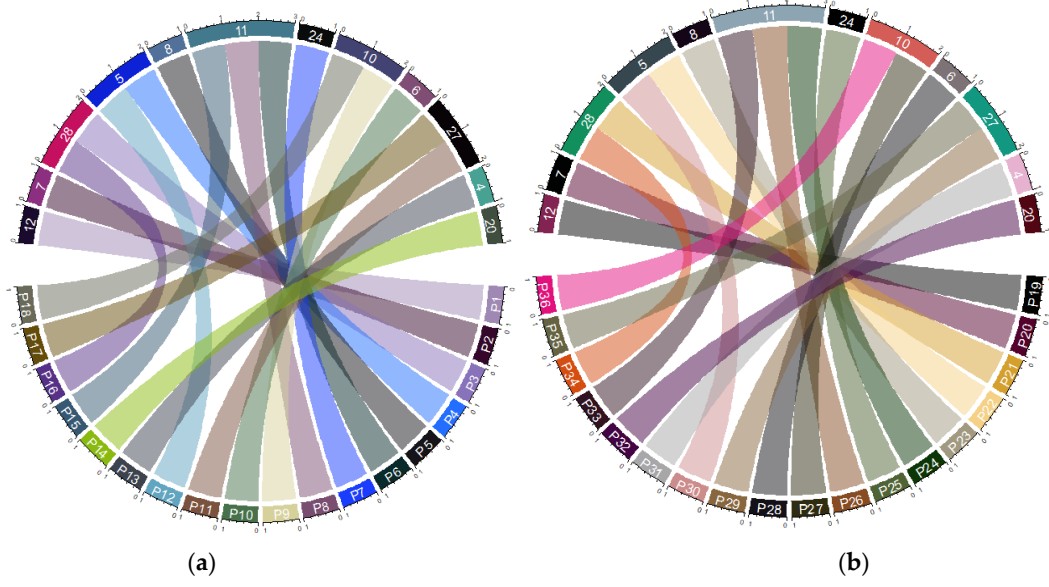

**Figure 2.** Sample reprezentation by age: (**a**) Prader–Willi sample; (**b**) control sample. Source: authors' contribution in R version 3.6.3.

### 2.2. Research Methodology—Models of Analysis

This research relies on a complex modelling endeavor comprising GGMs for the network analysis and SEM deployed to capture the patterns and overall linkages between 29 cephalometric measurement units performed on the two samples (Prader–Willi and Control).

The network analysis is based on GGMs which are innovative exploratory research tools, very useful to infer the connections/dependencies between variables (particularly biological variables and genes). For this research, GGMs were configured and processed through the extended Bayesian information criteria (EBIC) with graphical (g) least absolute shrinkage and selection operator (lasso) (EBICglasso) and partial correlation (Pcor). A GGM is a graph in which all random variables are continuous and jointly Gaussian [17,18] and it is based on conditional independence, respectively if $\Omega = \{\omega_{v1,v2}\}$, two variables v1 and v2 are conditionally independent if $\omega_{v1,v2} = 0$, namely there are 0 entries of the precision matrix $\Omega = \Sigma^{-1}$. $\Sigma$ is the positive definite covariance matrix and $\Omega$ is the precision matrix of the distribution, defined as the inverse of $\Sigma$.

If $\Sigma$ is positive definite, distribution has density on $f(x|\xi, \Sigma) = (2\pi)^{-d/2}(\det\Omega)^{1/2}e^{-(x-\xi)^T\Omega(x-\xi)/2}$. The sample covariance matrix is represented by $\bar{\Sigma} = \frac{1}{n-1}\sum_{i=1}^{n}(x_i - \xi)(x_i - \xi)^T$ [17,18].

GGMs "entail an undirected network of partial correlation coefficients (both positive and negative). They are graphically reflected through the absolute strengths (width and saturation of the edges between nodes), thus being a network model of conditional associations and avoiding spurious correlation".

The partial correlation (pcor) determined in the GGMs can be calculated as in Equation (1) [19]:

$$r_{xy\cdot z} = \frac{r_{xy} - r_{xz}r_{yz}}{\sqrt{1 - r_{xz}^2}\sqrt{1 - r_{yz}^2}} \tag{1}$$

where: *r* represents the correlation degree.

In our GGM networks, positive partial correlations are generally visualized with blue edges and negative partial correlations with red edges.

GGMs are complemented in our research with another modern technique and approach to modelling longitudinal data, namely SEM. Both techniques "imply a variance–covariance matrix, aiming to identify how variables are related to each other, namely the direct and indirect effects of one variable on another, having their origin in path analysis" [19].

SEM are modern research instruments that allowed us to identify and assess the latent interconnections between numerous cephalometric measurements specific for the Prader–Willi syndrome diagnostic, so as to infer the positive and negative influences in this regard. The general configuration of the SEM models designed in our research is presented in Figure 3.

"Going beyond the classical linear regression analyses, SEM examines the causal relationships among variables, while controlling simultaneously for measurement error as a greatest advantage in empirical researches". SEM allowed us to determine the degree of correlation (path coefficients) that capture the importance of a certain path of influence from cause to effect [20].

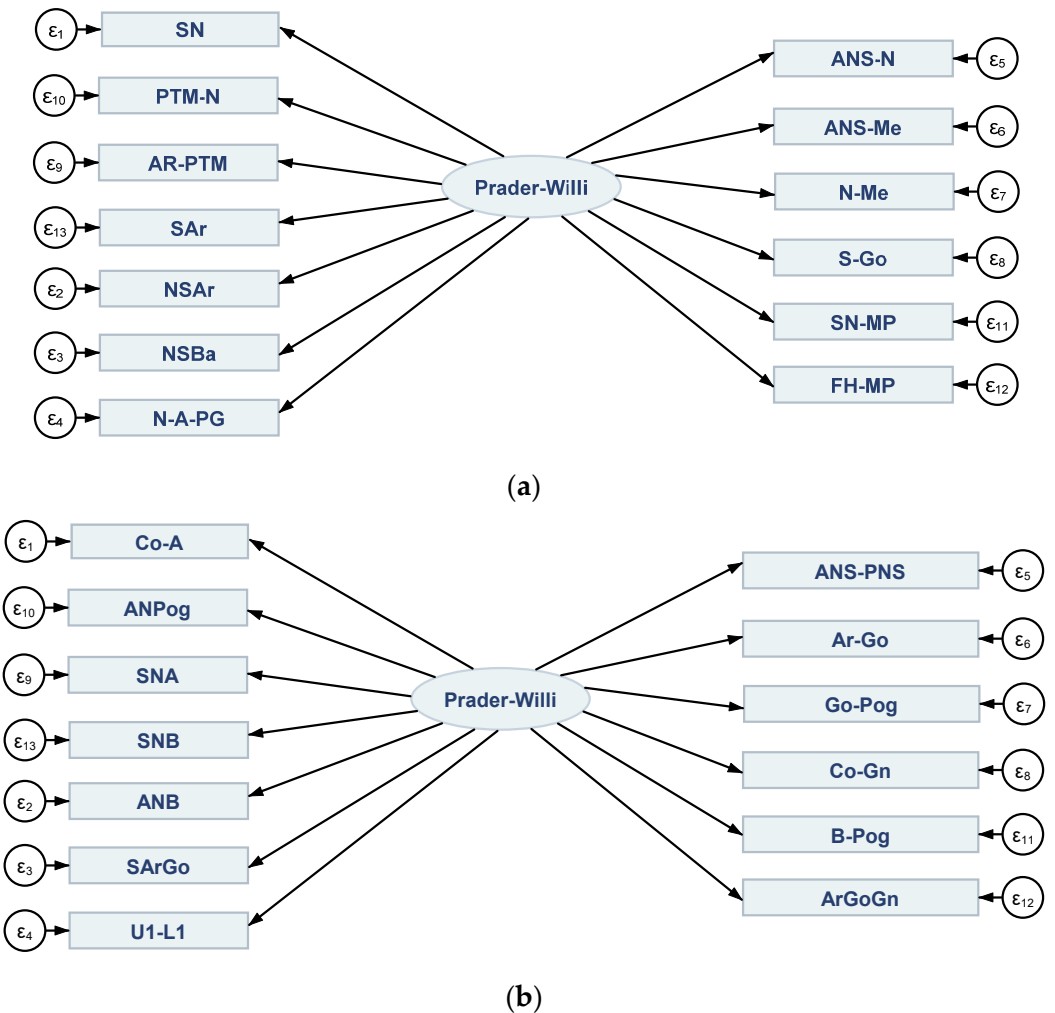

**Figure 3.** General configuration of the structural equation models (SEM): (**a**) first set of indicators (cephalometric measures); (**b**) second set of indicators (cephalometric measures). Source: authors' configuration in Stata 16.

## 3. Results

*3.1. Results of the Gaussian Graphical Models (GGMs) Entailing the Connections and Correlations between Considered Cephalometric Measures in Both Prader–Willi Syndrome (PWS) Group and Control Sample*

In the Prader–Willi sample, the first set of GGMs configured through the extended Bayesian information criteria with graphical lasso and partial correlations (Figure 4) entail dominant positive connections between the SN line (anterior cranial base) and posterior facial height (S-Go) and cranial base flexure angle (NSBa). SN "is often used by orthodontists as a reference line for assessment of dentofacial deformities [21], being a suitable assessor for exact facial measurements, that are also major criteria in determining the Prader–Willi diagnostic. Strong positive connections are also grasped between S-Go and SAr, NSAr, SN-MP, FH-MP (direct linkages between the posterior facial height and posterior cranial base, cranial base angle, the mandibular plane and its inclination to FH) and negative ones between SAr (posterior cranial base) and SN-MP, FH-MP, SN (Sella–Nasion line, namely the anterior cranial base). At the same time, N-Me, ANS-Me and ANS-N (total, lower and upper anterior facial height) also represent basic credentials that stand out in the Prader–Willi sample, in this pre-definite framework of relationships/measurements.

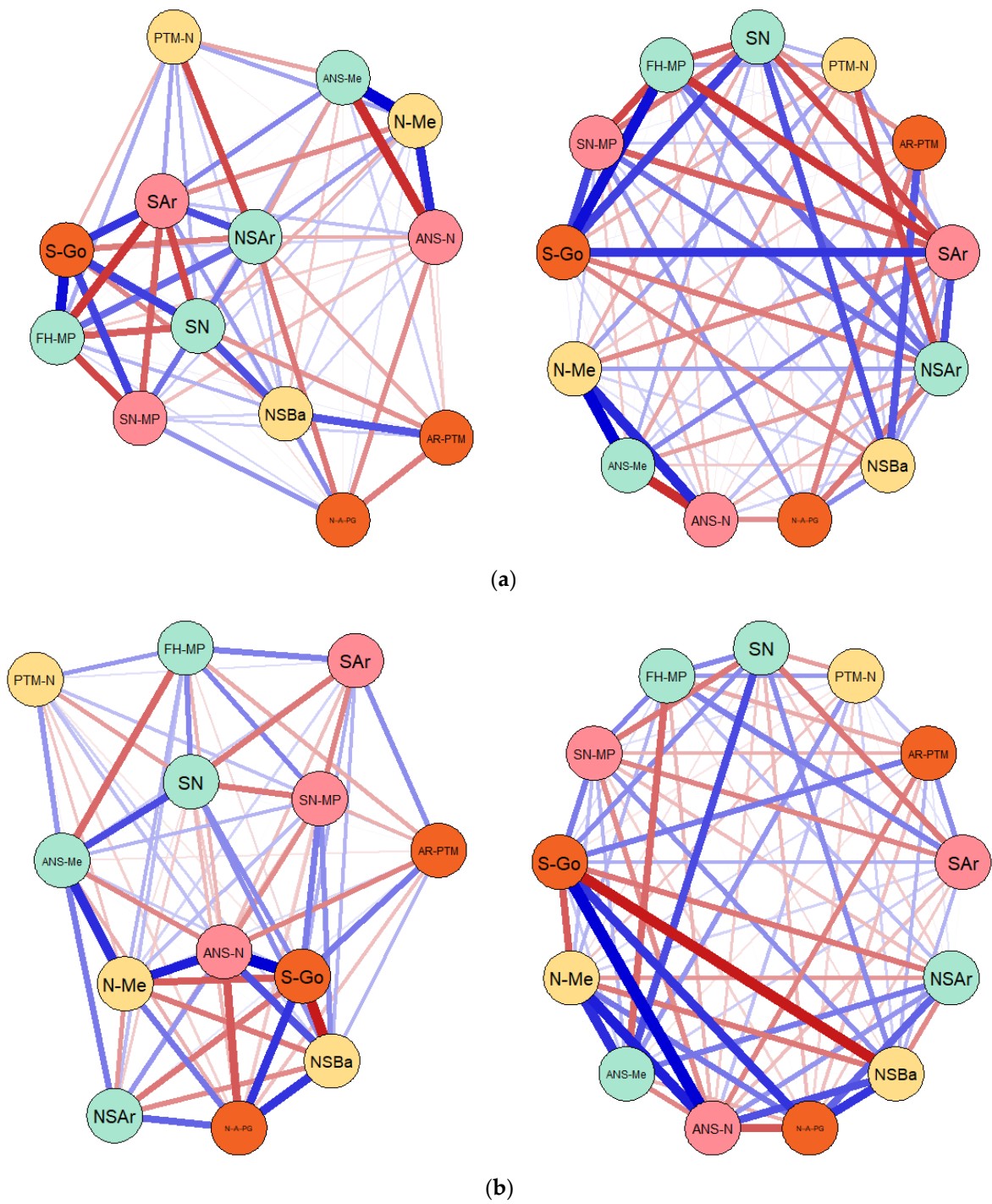

**Figure 4.** Results of the Gaussian graphical models (GGMs), processed through extended Bayesian information criteria with graphical least absolute shrinkage and selection operator (lasso) (EBICglasso) and partial correlation (Pcor): (**a**) Prader–Willi sample—first set of indicators/measurements; (**b**) control sample—first set of indicators/measurements. Source: authors' configuration in Stata 16.

When we have considered the control sample, after configuring the GGMs, we can observe that there are very intense linkages between S-Go (posterior facial height) and ANS-N (upper anterior facial height) (strongly positive), N-Me (total anterior facial height) (negative), N-A-PG (angle of facial convexity) (positive) and NSBa (cranial base flexure angle) (strongly negative). As in the Prader–Willi sample, N-Me is dominant and positively correlated with ANS-Me and ANS-N, the latter two being placed in an indirect connection.

SN is also strongly and positively connected with ANS-Me and inversely connected with SAr and SN-MP.

In both samples, facial characteristic (particularly the total posterior facial height in the Prader–Willi sample) represent an important credential in setting the diagnostic. In the Prader–Willi, the posterior facial height should be considered in strong dependency with the mandibular plane, posterior cranial base, cranial base angle (saddle angle) and Sella–Nasion line, since our Gaussian graphical models have evidenced strong edges and partial correlations between these credentials. These parameters tend to have reduced dimensions compared to the control sample. In our study group with Prader–Willi patients we found a reduced cranial base angle (NSAr = 118.15 compared to 123.07 in the control sample) that is associated with the brachiofacial model [22], which is the characteristic face pattern for Prader–Willi patients.

The second set of GGMs entail very strong and intense linkages between the second set of cephalometric measurements in case of the control sample (Figure 5a), while for the Prader–Willi sample these patterns are diminished. Very strong connections in the Prader–Willi sample stand out between A point to B point angle (ANB), Sella–Nasion to A point angle (SNA), Sella–Nasion to B point angle (SNB) and sagittal jaw relationship (ANPog). Hence, sagittal skeletal patterns (captured through ANB) tends to prevail in this sample, the relationship between the maxilla and mandible being very important. Also, the ANB is positively correlated with SNA and ANPog, respectively negatively correlated with SNB. In our Prader–Willi sample, the SNA has an average value of 83.85 (minimum 78.95 and maximum 86.52, hence the tendency in Prader–Willi syndrome patients is towards maxillary prognathism), compared to the control sample where SNA has an average value of 81.83 (very few variations across sample, values ranging from a minimum of 81.12 to a maximum of 82.85, hence a normal maxillary position). At the same time, the SNB registers an average value of 80.11 in the Prader–Willi sample and 79.16 in the control sample. The minimum value of SNB in Prader–Willi sample goes to 75.79, thus reflecting a pattern of mandibular retrognathism compared to 77.23 (very close to the reference value of 78°) in the control sample. Also, in the control sample there is an indirect (negative) correlation between SNA and ANS-PNS (anterior-posterior nasal spine/palatal plane), while in the Prader–Willi sample this correlation does not seem to exist.

At the same time, in the Prader–Willi sample, we could also observe other interconnections, namely: U1–L1 (dental angular measurement, namely the interincisal angle) is another credential that stands out, being strongly and positively partially correlated with Co-A (maxilla length) and SAr-Go (articular angle), and negatively partially correlated with B-Pog (chin depth) and ArGoGn (the Gonial angle), but with a lower intensity of the connection. Hence, the Prader–Willi syndrome caused dentofacial abnormalities that are associated also with a modified maxillary length. Co-A and SAr-Go are strongly and negatively correlated in the Prader–Willi sample, while in the control sample these variables are strongly and positively correlated. There is also a very strong negative relationship/correlation between Co-A (maxilla length) and Go-Pog (mandibular body length) in the control sample, while in the Prader–Willi sample this relationship has faded.

### 3.2. Results of the Structural Equation Modelling (SEM) Conveying Direct, Indirect and Total Linkages between Cephalometric Characteristics in Both PWS Group and Control Sample

To further entail the importance of various facial-oral characteristics in determining the Prader–Willi syndrome diagnostic, in a case-control approach, we extended our research endeavor with a set of four SEMs. The results of the SEM models are presented in Figure 6a–d and detailed in the Appendix A, Tables A2 and A3, along with Tables A4 and A5 that capture the alpha Cronbach calculations (revealing robustness and a high reliability of the scale for all four SEM models) and Table A6 that captures the goodness of fit tests (showing that SEM models are well fitted and the results are robust).

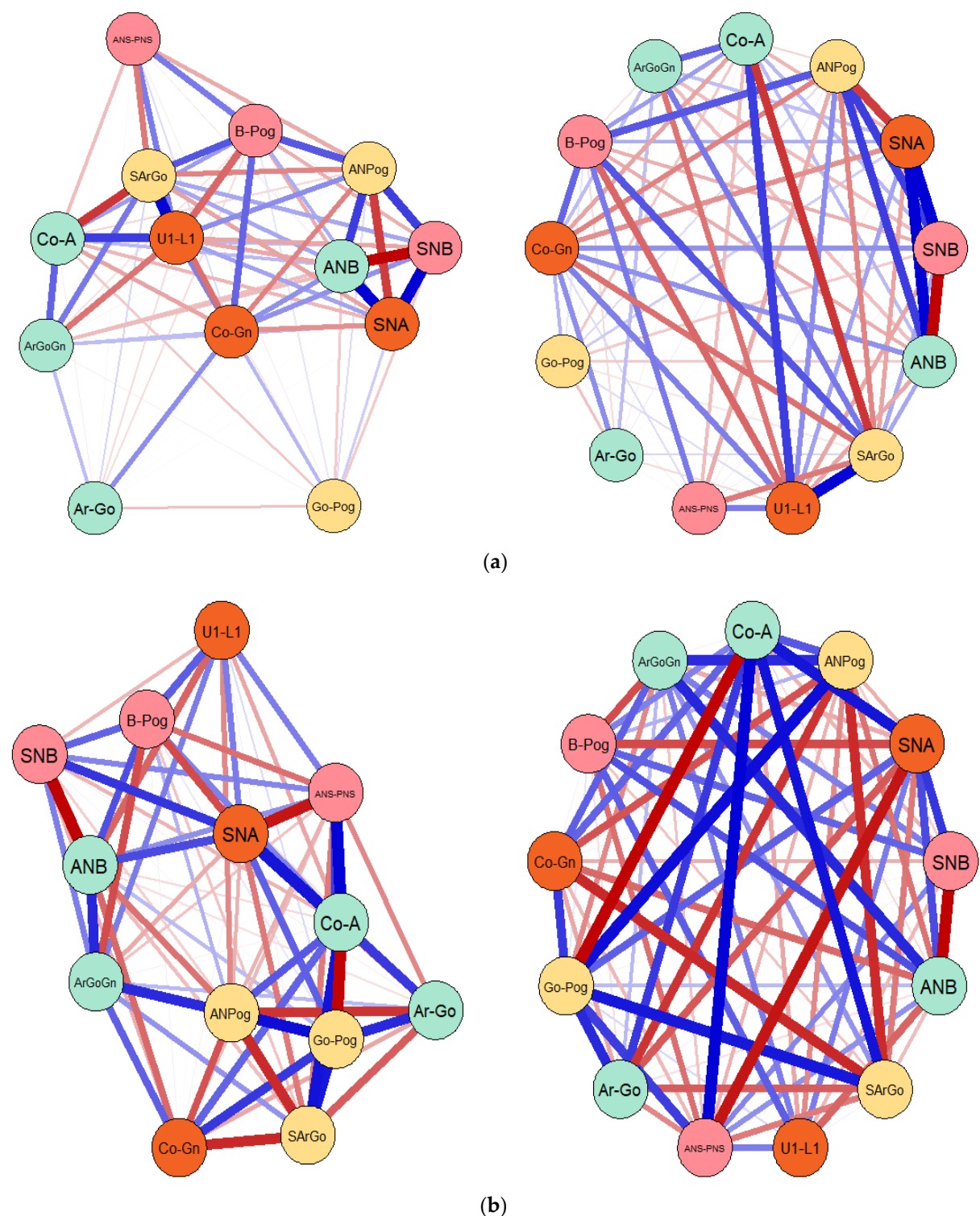

**Figure 5.** Results of Gaussian graphical models GGMs, processed through extended Bayesian information criteria with graphical least absolute shrinkage and selection operator (lasso) (EBICglasso) and partial correlation (Pcor): (**a**) Prader–Willi sample—second set of indicators/measurements; (**b**) control sample—second set of indicators/measurements. Source: authors' configuration in Stata 16.

As a modern technique, SEM allowed us to assess complex patterns of relationships among numerous measurements specific to the Prader–Willi genetic disorder, the latter being captured as a latent variable resulting from these credentials (Figure 6a,c). The same reasoning was applied for the control sample (Figure 6b,d) both being considered in a

comparative approach (case-control).

The results entail that the coefficients associated with SNA and SNB variables (measurements) are both negative (−0.75 and −0.84) and statistically significant in the case of the Prader–Willi sample, while for the control sample these coefficients are positive (0.35 and 0.75), being also very significant from a statistical point of view. These coefficients entail that an increase in the prevalence of PWS is associated with a significant change and variation in the SNA and SNB values.

At the same time, the coefficients associated with the facial height characteristics captured through ANS-N, ANS-Me, N-Me, S-Go are all positive in the control sample, while for the Prader–Willi group two of them are negative, mainly for ANS-N (anterior facial height) and N-Me (total anterior facial height). The negative coefficients entail that an increase in the probability of Prader–Willi syndrome is strongly associated with a reduced dimension of these parameters. The results are in line with Fields et al. [23] and reconfirm previous GGM results that have also entailed facial height as main parameter in determining the Prader–Willi syndrome diagnostic.

Negative coefficients, statistically significant, were obtained after processing the SEM models on the Prader–Willi sample also for the cranial base measurements, namely PTM-N, AR-PTM, SAr, NSAr and NSBa. These results highlight that an increase in the prevalence of PWS is associated with reduced/diminished values for the cranial base parameters.

Nevertheless, the coefficients associated with the palatal plane (ANS-PNS) and other three parameters reflecting mandibular ramus length (Ar-Go), mandibular body-lengths (Go-Pog) and the length of the mandibular base (Co-Gn) are positive and highly significant from a statistical point of view at the level of 0.1%, both in the PWS sample and the control group.

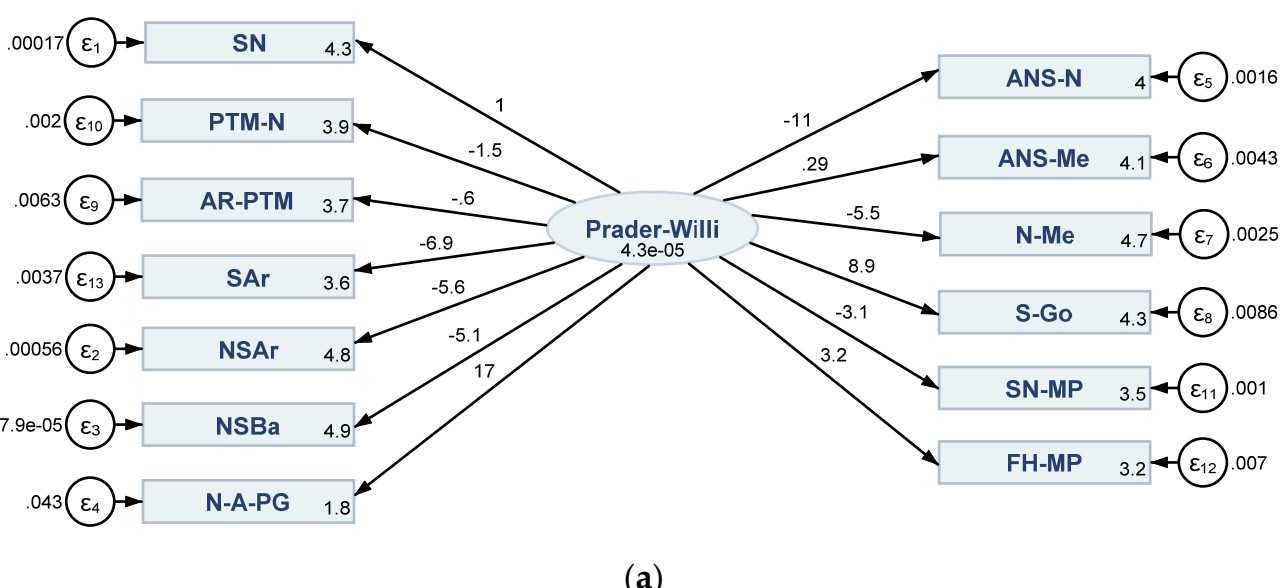

(**a**)

**Figure 6.** *Cont.*

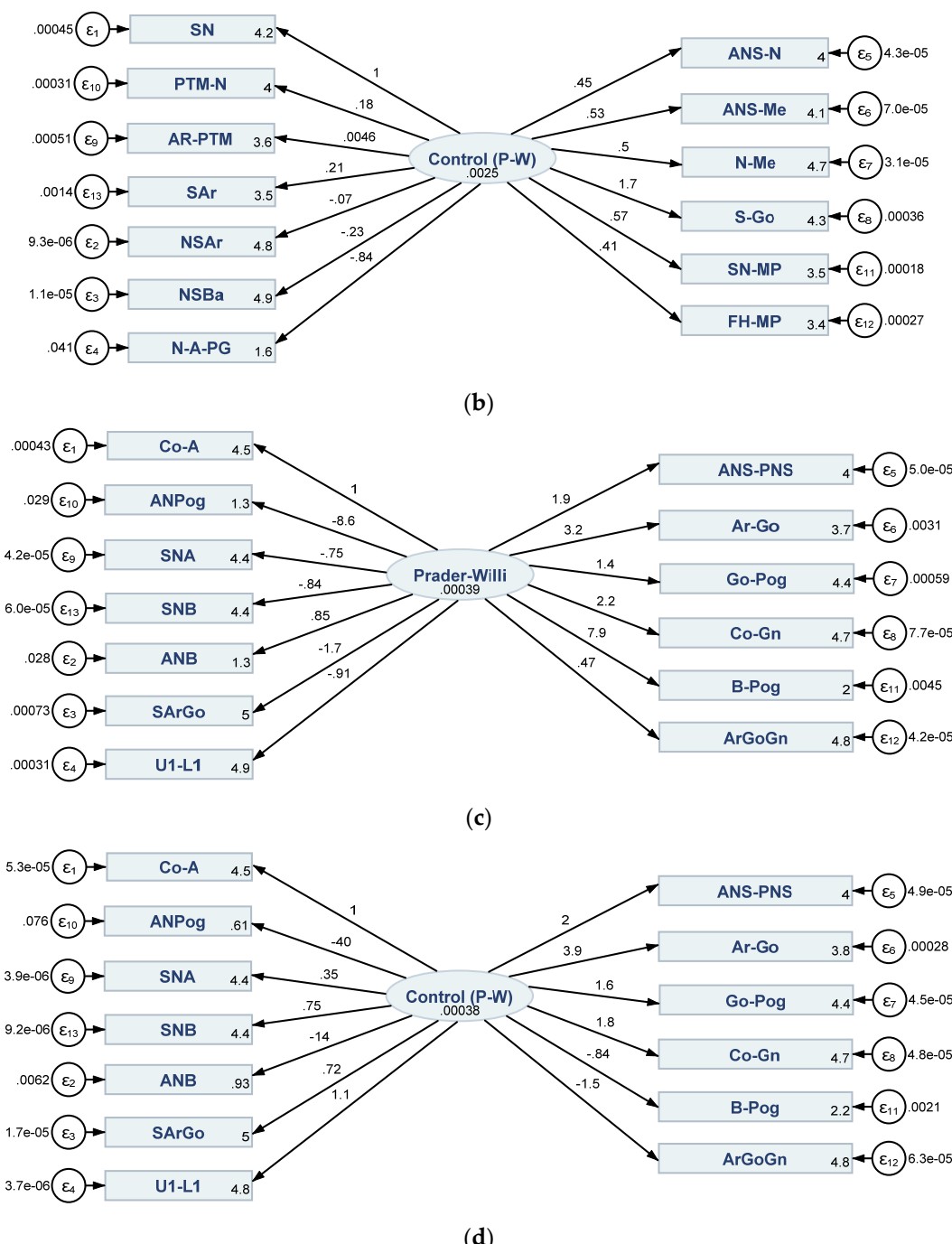

**Figure 6.** Results of the SEM models: (**a**) Prader–Willi sample—first set of indicators; (**b**) control sample—first set of indicators; (**c**) Prader–Willi sample—second set of indicators; (**d**) control sample—second set of indicators Source: authors' configuration in Stata 16.

## 4. Discussion

The early diagnosis of Prader–Willi syndrome is very important, especially during the neonatal period, in order to improve the treatment steps. Prader–Willi syndrome is a complex disorder with an involvement of the hypothalamus affecting the skeletal development, the endocrine function, and the intellect among other issues. Cranio-facial alterations have been present in the description of the syndrome, and the results of multiple studies stated them as a consequence of the disorder. The monitoring and management are crucial in these cases, as the complications can lead to an undesired outcome.

The facial morphology of the PWS patients showed reduced cranio-facial dimensions, affecting the growth of the maxilla and mandible [24]. Saeves et al. [24] in their study concluded that patients diagnosed with PWS developed a dysfunction of their oral motor function suggesting that an early interdisciplinary approach treatment is necessary in order to modify the maxillofacial growth pattern. Pearson et al. [25] were among the first that reported an obvious reduction of the mandibular length in patients with PWS. A study conducted by Hall et al. [26] that included 32 patients diagnosed with Prader–Willi syndrome showed that 40% of them had a reduction in the maxillofacial skeletal measurements. Schaedel et al. [14] concluded after the examination of 20 patients with PWS that a reduction is common in the total mandibular length, the body of the mandible, the length of the ramus, the maxillary length, the mid-facial and posterior facial height. The position of the maxillary bone in the sagittal plane was the subject of previous studies that reported an increased SNA angle in PWS patients [27].

Our results are in line with Schaedel et al. [14] and Belengeanu et al. [27] that have obtained similar reduced dimensions for the majority of parameters when compared to the control sample.

Also, after the cephalometric analysis that certified a reduction of the mandible, concerning the body and ramus, allowed several conclusions regarding the incidence of sleep apnea in patients with PWS [28].

The cephalometric analysis is the first step in acknowledging the alterations of the cranio-facial features, the measurements need to be correlated in order to evaluate the growth pattern and assign early treatment to avoid potential complications. Davidopoulou et al. [29] stated the fact that after the treatment with growth hormone the facial convexity decreased, the posterior facial height and mandibular length increased in time.

The first set of GGMs entailed the importance of posterior cranial base (SAr), posterior facial height (S-Go), the mandibular plane (SN-MP) and its inclination (FH-MP), and Sella–Nasion line (SN) as important fundamentals that stand out through their strong direct (positive) and indirect (negative) interconnections in the Prader–Willi sample. On the other hand, in the control sample, S-Go is strongly and negatively connected with NSBa (cranial base flexure angle) and positively connected with the upper anterior facial height (ANS-N) and the angle of facial convexity (N-A-PG).

The second set of Gaussian models reveal different patterns of relationship between specific cranial, facial and oral credentials in the two samples considered for analysis. For the Prader–Willi sample, the GGMs processed through EBICglasso and partial correlations reveal that ANB, SNA, SNB and AnPog, Co-A, Co-Gn and U1–L1 are essential credentials with strong interlinkages amongst them (both positive and negative). In the control sample, however, all the connections are strapping and robust, being hard to distinguish certain relationships as being more important than the others.

Facial heights are outlines for the concept of facial harmony which implies a certain ratio between these measurements (Fields et al., 1984). In our research, for the Prader–Willi sample all the facial heights (ANS-N, ANS-Me, N-Me, S-Go) accounted reduced average dimensions compared to the control group (in the Prader Willi group: ANS-N = 53.02, ANS-Me = 58.04, N-Me = 111.46, S-Go = 71.94; in the control sample: ANS-N = 54.25, ANS-Me = 61.36, N-Me = 115.50, S-Go = 73.60). Moreover, all of these facial height credentials, and particularly the posterior facial height (S-Go) strongly and partially correlated with the cranial base measurements.

Despite its significance, research on Prader–Willi genetic disorder, as well as on craniofacial syndromes and various groups of deformities, still needs to be strengthened with comprehensive detailed assessments on the role played by each individual feature/personal characteristic and cephalometric measurement in setting up the diagnostic. To the best of our knowledge, the literature still lacks additional empirical evidence brought by advanced researches in the field to attest to the decisive importance of cephalometric measures in PWS diagnostic. Hence, our paper seeks to fill in these gaps and provide accurate, robust empirical evidences on the role played by cephalometric characteristics and associated

patterns in the PWS diagnostic, but also even further as regards specific treatment procedures. It further presents the advantage of embedding various modern methods and techniques, like GGMs and SEM, less considered in medical research but with a tremendous potential [18,20], that allow a comprehensive approach on the interlinkages between all specific features of the PWS in a predefined medical setting, while also controlling for misspecifications, thus ensuring accurate ways to address these research questions.

The use of cephalometric evaluation was analyzed in several studies including patients with microdeletion syndrome and is consider to be valuable to clinicians, including dental practitioners [30,31]. For the microdeletion syndromes exhibiting heterogeneous phenotypes, the diagnosis can be difficult and the patients remain undiagnosed until a later age. By using modern methods and techniques, like GGMs and SEM, new insights into the recognition pattern of genetic syndromes can be acquired and might help in the development of algorithms for facial analysis in order to assist the clinical evaluation.

## 5. Conclusions

This research was conducted in the open conversation on the fundamental cranio-facial credentials and their interlinkages as essential milestones in determining the Prader–Willi syndrome diagnostic. Therefore, we have aimed to cover the existing gaps and bring new evidence from Gaussian graphical models (GGMs) and structural equation modeling (SEM) by integrating the variables related to the cranio-facial alterations found in Prader–Willi syndrome and to capture the patterns and overall linkages between the measurement units from our two groups that are included in the study. The research, therefore, contributes to the existing literature and provides new empirical evidence and a comprehensive assessment of Prader–Willi craniofacial patterns and characteristics. Main findings of this research may be considered as guidelines in the field of craniofacial diseases, particularly in what concerns the role played by specific cephalometric measurements and associated patterns/features in setting up the diagnostic, as most of these conditions have distinct characteristics and craniofacial anomalies are often associated with a variety of genetic syndromes.

The research endeavor accounts as limitation the relatively reduced size of the sample, an issue that is however balanced by the fact that the analysis relies on an extremely rare genetic disorder thus making it difficult to cover for a larger sample. Future research directions target an increased number of Prader–Willi syndrome subjects and an advanced analysis on sub-samples determined according to the age of the patients, namely a separate focus on children and adults, in order to better capture the differentials amongst them.

**Author Contributions:** Conceptualization, A.V.I., L.-C.R., G.G.N., N.I.A., A.R. and G.M.; methodology, G.G.N. and G.M.; software, G.G.N.; validation, G.G.N., L.-C.R., G.M., A.V.I., M.P., S.S.F., N.I.A. and E.B.; formal analysis, G.G.N. and E.B.; investigation, L.-C.R., G.G.N., N.I.A., A.V.I.; resources, L.-C.R., G.G.N., N.I.A.; data curation, L.-C.R., M.P., S.S.F. and N.I.A.; writing—original draft preparation, G.G.N., L.-C.R., N.I.A., A.R.; writing—review and editing, G.G.N., L.-C.R., G.M., M.P., N.I.A., A.R.; visualization, G.G.N., C.R., N.I.A., M.P. and A.V.I.; supervision, G.G.N., L.-C.R., N.I.A. and A.V.I. All authors have read and agreed to the published version of the manuscript.

**Funding:** This research received no external funding.

**Institutional Review Board Statement:** Ethical review and approval were waived for this study, due to the fact that it did not involve any inference of any nature on patients, being an observational study.

**Informed Consent Statement:** Patient consent was waived because it did not involve any personal information.

**Data Availability Statement:** The dataset can be provided by the authors upon request.

**Conflicts of Interest:** The authors declare no conflict of interest.

## Appendix A

**Table A1.** Detailed description of the variables (specific measurements) used in the empirical analysis.

| Acronym | Variable/Measure—Detailed Description |
|---------|---------------------------------------|
| SN | SN represents Sella-Nasion line |
| PTM-N | PTM-N represents anterior cranial base length |
| AR-PTM | AR-PTM represents posterior cranial base length |
| SAr | SAr represents posterior cranial base |
| NSAr | NSAr represents saddle angle |
| NSBa | NSBa represents cranial base flexure angle |
| N-A-PG | N-A-PG represents the angle of facial convexity |
| ANS-N | ANS-N (UAFH) represents upper anterior facial height |
| ANS-Me | ANS-Me (LAFH) represents lower anterior facial height |
| N-Me | N-Me represents total anterior facial height |
| S-Go | S-Go represents total posterior facial height |
| SN-MP | SN-MP represents mandibular plane |
| FH-MP | FH-MP represents the inclination of mandibular plane to FH |
| Co-A | Co-A represents maxilla length |
| ANPog | ANPog represents sagittal jaw relationship |
| SNA | SNA represents Sella-Nasion to A Point Angle |
| SNB | SNB represents Sella-Nasion to B Point Angle |
| ANB | ANB represents A point to B Point Angle |
| SArGo | SArGo represents articular angle |
| U1-L1 | U1-L1 represents the interincisal angle |
| ANS-PNS | ANS-PNS represents palatal plane |
| Ar-Go | Ar-Go represents mandibular ramus length |
| Go-Pog | Go-Pog represents mandibular body length |
| Co-Gn | Co-Gn length of mandibular base |
| B-Pog | B-Pog represents chin depth |
| ArGoGn | ArGoGn represents Gonial angle |

Source: Authors' configuration.

**Table A2.** Detailed SEM results associated with Figure 6a,b.

| | (1) | (2) |
|---|---|---|
| | **Prader-Willi** | Control |
| **SN** | | |
| Prader–Willi | 1 | 1 |
| | (.) | (.) |
| _cons | 4.328 *** | 4.250 *** |
| | (0.00348) | (0.0128) |
| **NSAr** | | |
| Prader–Willi | −5.631 | −0.0695 *** |
| | (2.895) | (0.0161) |
| _cons | 4.771 *** | 4.813 *** |
| | (0.0104) | (0.00109) |

**Table A2.** *Cont.*

|  | (1) | (2) |
| --- | --- | --- |
|  | **Prader-Willi** | **Control** |
| NSBa |  |  |
| Prader–Willi | −5.072 * | −0.229 *** |
|  | (2.561) | (0.0278) |
| _cons | 4.851 *** | 4.873 *** |
|  | (0.00814) | (0.00282) |
| N_A_PG |  |  |
| Prader–Willi | 17.00 | −0.837 |
|  | (11.29) | (0.968) |
| _cons | 1.838 *** | 1.628 *** |
|  | (0.0556) | (0.0489) |
| ANS_N |  |  |
| Prader–Willi | −10.61 * | 0.447 *** |
|  | (5.411) | (0.0549) |
| _cons | 3.968 *** | 3.993 *** |
|  | (0.0190) | (0.00550) |
| ANS_Me |  |  |
| Prader–Willi | 0.288 | 0.528 *** |
|  | (2.466) | (0.0665) |
| _cons | 4.059 *** | 4.117 *** |
|  | (0.0154) | (0.00654) |
| N_Me |  |  |
| Prader–Willi | −5.479 | 0.505 *** |
|  | (3.241) | (0.0575) |
| _cons | 4.712 *** | 4.749 *** |
|  | (0.0146) | (0.00610) |
| S_Go |  |  |
| Prader–Willi | 8.877 | 1.651 *** |
|  | (5.565) | (0.189) |
| _cons | 4.270 *** | 4.295 *** |
|  | (0.0259) | (0.0200) |
| AR-PTM |  |  |
| Prader–Willi | −0.595 | 0.00457 |
|  | (3.064) | (0.107) |
| _cons | 3.681 *** | 3.616 *** |
|  | (0.0188) | (0.00530) |
| PTM-N |  |  |
| Prader–Willi | −1.498 | 0.177 * |
|  | (1.856) | (0.0851) |
| _cons | 3.938 *** | 3.953 *** |
|  | (0.0108) | (0.00464) |
| SN-MP |  |  |
| Prader–Willi | −3.100 | 0.572 *** |
|  | (1.917) | (0.0854) |
| _cons | 3.538 *** | 3.525 *** |
|  | (0.00894) | (0.00744) |
| FH-MP |  |  |
| Prader–Willi | 3.219 | 0.405 *** |
|  | (3.475) | (0.0874) |
| _cons | 3.237 *** | 3.360 *** |
|  | (0.0204) | (0.00614) |

**Table A2.** *Cont.*

|  | (1) | (2) |
|---|---|---|
|  | **Prader-Willi** | **Control** |
| **SAr** |  |  |
| Prader–Willi | −6.942 (4.059) | 0.206 (0.179) |
| _cons | 3.592 *** (0.0179) | 3.503 *** (0.00918) |
| **/** |  |  |
| var(e.SN) | 0.000175 ** (0.0000595) | 0.000452 ** (0.000163) |
| var(e.NSAr) | 0.000557 ** (0.000216) | 0.00000926 ** (0.00000318) |
| var(e.NSBa) | 0.0000789 (0.0000843) | 0.0000107 * (0.00000417) |
| var(e.N-A-PG) | 0.0432 ** (0.0147) | 0.0413 ** (0.0138) |
| var(e.ANS-N) | 0.00160 * (0.000657) | 0.0000430 * (0.0000175) |
| var(e.ANS-Me) | 0.00427 ** (0.00142) | 0.0000703 * (0.0000304) |
| var(e.N-Me) | 0.00254 ** (0.000886) | 0.0000307 (0.0000163) |
| var(e.S-Go) | 0.00863 ** (0.00294) | 0.000357 (0.000191) |
| var(e. AR-PTM) | 0.00635 ** (0.00212) | 0.000506 ** (0.000169) |
| var(e. PTM-N) | 0.00200 ** (0.000670) | 0.000308 ** (0.000103) |
| var(e.SN-MP) | 0.00102 ** (0.000350) | 0.000176 ** (0.0000624) |
| var(e.FH-MP) | 0.00702 ** (0.00235) | 0.000266 ** (0.0000906) |
| var(e.SAr) | 0.00369 ** (0.00128) | 0.00141 ** (0.000470) |
| var(Prader–Willi) | 0.0000433 (0.0000450) | 0.00251 * (0.000977) |
| *N* | 18 | 18 |

Note: Standard errors in parentheses, * $p < 0.05$, ** $p < 0.01$, *** $p < 0.001$. Source: Authors' research in Stata 16.

**Table A3.** Detailed SEM results associated with Figure 6c,d.

|  | (1) | (2) |
|---|---|---|
|  | **Prader–Willi** | **Control** |
| **Co-A** |  |  |
| Prader–Willi | 1 (.) | 1 (.) |
| _cons | 4.493 *** (0.00694) | 4.457 *** (0.00491) |
| **ANB** |  |  |
| Prader–Willi | 0.847 (2.110) | −13.56 *** (1.534) |
| _cons | 1.312 *** (0.0410) | 0.933 *** (0.0651) |

**Table A3.** *Cont.*

|  | (1) | (2) |
| --- | --- | --- |
|  | **Prader–Willi** | **Control** |
| SArGo |  |  |
| Prader–Willi | −1.750 ** | 0.723 *** |
|  | (0.558) | (0.0808) |
| _cons | 4.985 *** | 4.962 *** |
|  | (0.0106) | (0.00346) |
| ANS-PNS |  |  |
| Prader–Willi | 1.880 *** | 2.012 *** |
|  | (0.496) | (0.197) |
| _cons | 4.017 *** | 4.010 *** |
|  | (0.00917) | (0.00940) |
| Ar-Go |  |  |
| Prader–Willi | 3.209 ** | 3.873 *** |
|  | (1.080) | (0.397) |
| _cons | 3.659 *** | 3.785 *** |
|  | (0.0205) | (0.0182) |
| Go-Pog |  |  |
| Prader–Willi | 1.448 ** | 1.632 *** |
|  | (0.487) | (0.165) |
| _cons | 4.394 *** | 4.409 *** |
|  | (0.00911) | (0.00767) |
| Co-Gn |  |  |
| Prader–Willi | 2.158 *** | 1.770 *** |
|  | (0.571) | (0.177) |
| _cons | 4.713 *** | 4.724 *** |
|  | (0.0106) | (0.00830) |
| SNA |  |  |
| Prader–Willi | −0.752 *** | 0.353 *** |
|  | (0.209) | (0.0393) |
| _cons | 4.432 *** | 4.405 *** |
|  | (0.00393) | (0.00169) |
| ANPog |  |  |
| Prader–Willi | −8.604 ** | −40.00 *** |
|  | (3.067) | (4.861) |
| _cons | 1.324 *** | 0.609 ** |
|  | (0.0583) | (0.195) |
| B-Pog |  |  |
| Prader–Willi | 7.940 *** | −0.836 |
|  | (2.212) | (0.559) |
| _cons | 1.984 *** | 2.158 *** |
|  | (0.0413) | (0.0115) |
| ArGoGn |  |  |
| Prader–Willi | 0.467 ** | −1.541 *** |
|  | (0.144) | (0.167) |
| _cons | 4.822 *** | 4.845 *** |
|  | (0.00274) | (0.00733) |
| SNB |  |  |
| Prader–Willi | −0.839 *** | 0.753 *** |
|  | (0.237) | (0.0760) |
| _cons | 4.387 *** | 4.371 *** |
|  | (0.00444) | (0.00354) |

**Table A3.** *Cont.*

|  | (1) | (2) |
|---|---|---|
|  | **Prader–Willi** | **Control** |
| U1–L1 |  |  |
| Prader–Willi | −0.906 ** | 1.139 *** |
|  | (0.317) | (0.103) |
| _cons | 4.938 *** | 4.839 *** |
|  | (0.00609) | (0.00526) |
| / |  |  |
| var(e.Co-A) | 0.000430 ** | 0.0000532 ** |
|  | (0.000151) | (0.0000185) |
| var(e.ANB) | 0.0282 ** | 0.00624 ** |
|  | (0.00969) | (0.00217) |
| var(e.SArGo) | 0.000734 ** | 0.0000168 ** |
|  | (0.000259) | (0.00000586) |
| var(e.U1-L1) | 0.000310 ** | 0.00000370 |
|  | (0.000110) | (0.00000225) |
| var(e.ANS-PNS) | 0.0000504 | 0.0000495 ** |
|  | (0.0000315) | (0.0000191) |
| var(e.Ar-Go) | 0.00314 ** | 0.000281 ** |
|  | (0.00110) | (0.000102) |
| var(e.Go-Pog) | 0.000592 ** | 0.0000448 ** |
|  | (0.000211) | (0.0000162) |
| var(e.Co-Gn) | 0.0000772 | 0.0000478 ** |
|  | (0.0000443) | (0.0000174) |
| var(e.SNA) | 0.0000418 * | 0.00000390 ** |
|  | (0.0000171) | (0.00000136) |
| var(e.ANPog) | 0.0288 ** | 0.0759 ** |
|  | (0.0100) | (0.0261) |
| var(e.B_Pog) | 0.00445 ** | 0.00210 ** |
|  | (0.00168) | (0.000701) |
| var(e.ArGoGn) | 0.0000422 ** | 0.0000629 ** |
|  | (0.0000156) | (0.0000221) |
| var(e.SNB) | 0.0000605 ** | 0.00000918 ** |
|  | (0.0000224) | (0.00000334) |
| var(Prader–Willi) | 0.000390 | 0.000381 ** |
|  | (0.000242) | (0.000144) |
| N | 18 | 18 |

Note: Standard errors in parentheses, * $p < 0.05$, ** $p < 0.01$, *** $p < 0.001$. Source: Authors' research in Stata 16.

**Table A4.** Alpha Cronbach calculations associated with the SEM models presented in Figure 6a,b.

| | | SEM 1 (Figure 6a) | | | SEM 2 (Figure 6b) | | |
|---|---|---|---|---|---|---|---|
| Item | Obs | Sign | Item-Test Correlation | Alpha | Sign | Item-Test Correlation | Alpha |
| SN | 18 | − | 0.4909 | 0.8009 | + | 0.8890 | 0.9198 |
| NSAr | 18 | + | 0.7462 | 0.7754 | − | 0.7974 | 0.9237 |
| NSBa | 18 | + | 0.8229 | 0.7668 | − | 0.9180 | 0.9186 |
| N-A-PG | 18 | − | 0.5083 | 0.7993 | − | 0.3205 | 0.9412 |
| ANS-N | 18 | + | 0.9083 | 0.7567 | + | 0.8920 | 0.9197 |
| ANS-Me | 18 | + | 0.2839 | 0.8188 | + | 0.9209 | 0.184 |
| N-Me | 18 | + | 0.7710 | 0.7727 | + | 0.9171 | 0.9186 |
| S-Go | 18 | − | 0.5724 | 0.7932 | + | 0.9399 | 0.9176 |
| AR-PTM | 18 | − | 0.2849 | 0.8187 | + | 0.2284 | 0.9442 |
| PTM-N | 18 | + | 0.2532 | 0.8212 | + | 0.6200 | 0.9307 |
| SN-MP | 18 | + | 0.4036 | 0.8087 | + | 0.8800 | 0.9202 |
| FH-MP | 18 | − | 0.4201 | 0.8073 | + | 0.8199 | 0.9227 |
| SAr | 18 | + | 0.6876 | 0.7817 | + | 0.4809 | 0.9357 |
| Total scale | | | | 0.8081 | | | 0.9313 |

Source: Authors' research in Stata 16.

**Table A5.** Alpha Cronbach calculations associated with SEM models presented in Figure 6c,d.

| | | SEM 3 (Figure 6c) | | | SEM 4 (Figure 6d) | | |
|---|---|---|---|---|---|---|---|
| Item | Obs | Sign | Item-Test Correlation | Alpha | Sign | Item-Test Correlation | Alpha |
| Co-A | 18 | + | 0.7352 | 0.9114 | + | 0.9438 | 0.9846 |
| ANB | 18 | − | 0.1889 | 0.9335 | − | 0.9510 | 0.9844 |
| SArGo | 18 | − | 0.8377 | 0.9068 | + | 0.9667 | 0.9841 |
| U1-L1 | 18 | − | 0.7984 | 0.9087 | + | 0.9897 | 0.9836 |
| ANS-PNS | 18 | + | 0.8582 | 0.9057 | + | 0.9819 | 0.9838 |
| Ar-Go | 18 | + | 0.7641 | 0.9102 | + | 0.9740 | 0.9840 |
| Go-Pog | 18 | + | 0.4413 | 0.9235 | + | 0.9730 | 0.8840 |
| Co-Gn | 18 | + | 0.8389 | 0.9066 | + | 0.9799 | 0.9838 |
| SNA | 18 | − | 0.8441 | 0.9063 | + | 0.9687 | 0.9841 |
| ANPog | 18 | − | 0.7266 | 0.9087 | − | 0.9572 | 0.9843 |
| B-Pog | 18 | + | 0.6388 | 0.9150 | − | 0.4166 | 0.9945 |
| ArGoGn | 18 | + | 0.7140 | 0.9121 | − | 0.9584 | 0.9843 |
| SNB | 18 | − | 0.7913 | 0.9090 | + | 0.9744 | 0.9840 |
| Total scale | | | | 0.9187 | | | 0.9861 |

Source: Authors' research in Stata 16.

**Table A6.** "Goodness-of-fit tests" associated with the four SEM models presented in Figure 6a–d.

| | SEM 1 (Figure 6a) | SEM 2 (Figure 6b) | SEM 3 (Figure 6c) | SEM 4 (Figure 6d) |
|---|---|---|---|---|
| Likelihood ratio | | | | |
| "Model vs. saturated chi$^2$_ms (65)" | 312.611 | 143.313 | 306.380 | 165.709 |
| $p > $ chi$^2$ | 0.000 | 0.000 | 0.000 | 0.000 |
| "Baseline vs. saturated chi$^2$_bs (78)" | 390.637 | 408.698 | 535.035 | 706.875 |
| $p > $ chi$^2$ | 0.000 | 0.000 | 0.000 | 0.000 |
| Information criteria | | | | |
| "AIC (Akaike's information criterion)" | −642.733 | −1175.074 | −887.142 | −1295.027 |
| "BIC (Bayesian information criterion)" | −608.009 | −1140.350 | −854.646 | −1260.303 |
| Baseline comparison | | | | |
| "CFI (Comparative fit index)" | 0.208 | 0.769 | 0.472 | 0.840 |
| "TLI (Tucker–Lewis index)" | 0.050 | 0.723 | 0.366 | 0.808 |
| Size of residuals | | | | |
| "CD (Coefficient of determination)" | 0.957 | 0.989 | 0.987 | 0.997 |

Source: Authors' research in Stata 16.

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
