# Peer review of "An Observational Study on Cephalometric Characteristics and Patterns Associated with the Prader–Willi Syndrome: A Structural Equation Modelling and Network Approach"

_applsci, doi:10.3390/app11073177_

Round 1

Reviewer 1 Report

The study focuses on 18 subjects aged between 4 and 28 years who are genetically diagnosed with Prader-Willi syndrome, along with a healthy control group. Morphometry and cephalometry  are performed on the subjects to obtain cranio-facial parameters which are then statistically analyzed by Gaussian Graphical Models and Structural Equation Modeling. Hence, cranio-facial parameters of Prader-Willi syndrome patients are analyzed to determine patterns and overall linkages. The study presents new information on cranio-facial pattern and characteristics of Prader-Willi syndrome patients. The manuscript is interesting and presents new information. However, certain concerns need to be addressed.

  1. Abstract - needs to end with how the results impact the field, and how the methods can be applied further to improve the craniofacial field.
  2. Introduction - The last paragraph needs more elaboration regarding how the 2 techniques - Gaussian Graphical Models (GGMs) and Structural Equation Modeling (SEM) - are advantageous for this study, and if other craniofacial studies have benefited from them. 
  3. Introduction should end with a summary sentence of key results that links it to the Results section.
  4. Materials and Methods - A part of this section contains explanation on why GCMs and SEMs are used, some of that can be moved to the introduction to address concern #2.
  5. Results 3.1 and 3.2 - the titles of these sections need to be made specific to reflect what is being presented in these sections. Current titles are vague and general.
  6. Discussion - needs to include 1 or 2 sentences to describe how this study is advantageous over previous ones to address the concerned problem
  7. Conclusion - Potential impact on the field as a bigger picture pertaining to other craniofacial diseases need to be mentioned in 1 or 2 sentences.

Author Response

Dear Reviewer,

Thank you very much for the opportunity to reconsider the manuscript and to undertake revisions, marked with red colour into the document, by addressing the observations received! Thereby, we have performed several improvements that have added significant value to our research endeavour. We are very grateful for investing your time to analyse the paper and make very useful, encouraging and thoughtful comments and recommendations.

Thus, in accordance with the observations received, we have made the following revisions:

  1. As regards the first observation that „Abstract - needs to end with how the results impact the field, and how the methods can be applied further to improve the craniofacial field” – thank you very much! We have added an additional paragraph at the end of the abstract covering the issues outlined by the reviewer (lines 48-54).
  1. Regarding the second observation that „Introduction - The last paragraph needs more elaboration regarding how the 2 techniques - Gaussian Graphical Models (GGMs) and Structural Equation Modeling (SEM) - are advantageous for this study, and if other craniofacial studies have benefited from them” and the fourth observation that „Materials and Methods - A part of this section contains explanation on why GCMs and SEMs are used, some of that can be moved to the introduction to address concern #2” – we thank you very much for these suggestions and observations. We have moved several ideas from section 3 Materials and Methods to the Introduction, along with other mentions in this regard. Hence the Introduction has been significantly improved.  
  2. For the third observation that „Introduction should end with a summary sentence of key results that links it to the Results section” – thank you very much! We have added this summary sentence at the end of the introduction.
  3. As regards the observation that „Results 3.1 and 3.2 - the titles of these sections need to be made specific to reflect what is being presented in these sections. Current titles are vague and general” – we have reconfigured the titles of these two sections, thank you very much!
  4. Regarding the observation that „Discussion - needs to include 1 or 2 sentences to describe how this study is advantageous over previous ones to address the concerned problem” – thank you very much! We have added these sentences in Discussion section and further enhanced the innovative side of our paper.
  5. Regarding the observation that „Conclusion - Potential impact on the field as a bigger picture pertaining to other craniofacial diseases need to be mentioned in 1 or 2 sentences” – thank you very much! We have added these statements in the Conclusions section.

Reviewer 2 Report

This article concerns an observational study related to improving the diagnosis of Prader-Willi syndrome (PWS) and has a pure research character. In the Introduction the authors describe the medical background of PWS, while in the next chapter they already present the results of the obtained research. The research is related to structural equation modeling and network approach.

The disadvantage of the paper is the relatively small number of PWS subjects studied (only 18). For a research article, this is too few. Also, the topic of the paper is strongly medical in nature and is directed to a very narrow audience.

In addition to the above-mentioned comments, the paper does not raise my objections.

Author Response

Answer Reviewer 2

Dear Reviewer,

Thank you very much for the opportunity to reconsider the manuscript and to undertake revisions, marked with red colour into the document, by addressing the observations received! Thereby, we have performed several improvements that have added significant value to our research endeavour. We are very grateful for investing your time to analyse the paper and make very useful, encouraging and thoughtful comments and recommendations.

Thus, in accordance with the observations received, we have clearly stated in the Conclusions section the main limitation of our research endeavour related to a reduced size of the sample, an issue that is however balanced by the fact that the analysis relies on an extremely rare genetic disorder thus making it difficult to cover for a larger sample. Future research directions target an increased number of Prader-Willi syndrome subjects and an advanced analysis on sub-samples determined according to the age of the patients, namely a separate focus on children and adults, in order to better capture the differentials amongst them.

At the same time, we made several improvements throughout the paper, we enhanced the innovative side of the paper, extended and improved the Introduction section, along with Methods, Discussion and Conclusions sections.
